# Skin and Gut Microbiome in Hidradenitis Suppurativa: A Systematic Review

**DOI:** 10.3390/biomedicines11082277

**Published:** 2023-08-16

**Authors:** Edyta Lelonek, Dorra Bouazzi, Gregor B. E. Jemec, Jacek C. Szepietowski

**Affiliations:** 1Department of Dermatology, Venereology and Allergology, Wroclaw Medical University, 50-368 Wroclaw, Poland; jacek.szepietowski@umed.wroc.pl; 2Department of Dermatology, Zealand University Hospital, 4000 Roskilde, Denmark

**Keywords:** microbiome, gut, skin, hidradenitis suppurativa, review

## Abstract

Hidradenitis suppurativa (HS) is a chronic skin disease that significantly impairs the quality of life of affected individuals. The disease is characterized by persistent purulent lesions in specific anatomical areas, and its pathophysiology involves multiple factors, including inflammation, genetics, the microbiome, and environmental components. Recent research suggests a potential role for pathogenic bacteria in HS, highlighting the importance of the communication between the human host and the microbiome in maintaining homeostasis and immune system reactivity. However, the exact mechanisms underlying the gut–skin microbial interactions in HS remain unclear. This systematic review aims to examine the existing literature on the differences in skin and gut microbiome composition between HS patients and healthy controls. The review identifies methodological inconsistencies and calls for further research to elucidate the microbiome’s role in HS pathogenesis and to explore new therapeutic interventions. The review highlights the need for advancements in microbiome research methodologies, such as metataxonomics and metagenomics, to improve our understanding of the microbiota’s impact on health and disease.

## 1. Introduction

Hidradenitis suppurativa (HS) is a chronic autoinflammatory skin disease with persistently draining purulent lesions in intertriginous anatomical sites, such as the axilla, groin, and gluteal areas, leading to significantly impaired quality of life (QoL) due to the physical and psychological symptoms [1,2,3]. The pathophysiology of HS is multifactorial, with contributing factors including inflammation, genetic tendency, the microbiome, and environmental components. According to the classical postulate, the disease is initiated by follicular occlusion followed by the formation of inflammatory lesions and secondary bacterial infection [4].

New understanding of HS has begun to include a potential role for pathogenic bacteria. There is increasing evidence that the continuous crosstalk between humans and the microbiome is critical for establishment and maintenance of host homeostasis [5]. Alterations in microbiome composition could shift it towards dysbiosis—an abnormal state leading to modifications in immune system reactivity and subsequently to inflammatory disease development [6]. Emerging evidence supports the existence of the microbiome role in the pathogenesis of various cutaneous disorders, including psoriasis, atopic dermatitis, hidradenitis suppurativa, and acne, emphasizing the presence of communication axes between the gut and skin [7]. However, the exact mechanism underlying gut–skin microbial interactions has not yet been fully elucidated [8]. One of the therapeutic modalities in HS is the usage of antibiotics with known additional anti-inflammatory effects, for which the role of their antimicrobial action on the microbiome specifically is still unclear [9].

In recent years, advancements in the understanding of the complex relationship between the skin and gut microbiome have led to a thorough exploration of the microbiota composition. However, the association of HS with skin and/or gut dysbiosis is mainly based on limited studies with small numbers of patients involved. Thus, in this review, we scrutinize the existence of literature with regard to how the skin and gut microbiome in HS patients differs from that in healthy controls. Furthermore, the study aims to identify current methodological inconsistencies and outline directions for future research, which can contribute to the introduction of new strategic therapeutic interventions.

## 2. Methods

The systematic review was prospectively registered with PROSPERO (Registration Number CRD42022331681) and was conducted according to the Preferred Reporting Items for Systematic Reviews and Meta-Analyses (PRISMA) guidelines (Figure 1) [10]. The primary outcome measure of our study was a systematic review of previous studies investigating the skin and gut microbiome of patients with hidradenitis suppurativa. Discrepancies between investigators at every stage were thoroughly discussed by all authors until a consensus was reached.

### 2.1. Search Strategy

Studies were identified by searching electronic databases and scanning reference lists of articles. We searched four electronic databases from 1992 to May 2022: MEDLINE (via PubMed), Embase (via OvidSP), Web of Science Core Collection, and Google Scholar. To identify studies comparing skin and gut microbiome composition in patients with HS and normal healthy controls, the combination of the following keywords was included: “hidradenitis suppurativa”, “acne inversa”, “microbiome”, “microbiota”, “bacteriology”, “dysbiosis”, and “gut”. The Boolean operators used were “AND” and “OR”. Moreover, the reference sections of relevant articles were manually scanned for additional relevant studies. The database search was undertaken independently by two reviewers for each database. The titles and abstracts of articles retrieved by the search strategy were reviewed independently by two of the authors.

### 2.2. Inclusion/Exclusion Criteria

The inclusion criteria are as follows:Subjects aged 18 years or older;Human studies investigating the association between gut and/or skin microbiota and hidradenitis suppurativa;Articles published in English.

We excluded from the analysis articles not written in the English language, reviews, editorial letters, conference abstracts, expert opinions, and studies using animal models.

### 2.3. Study Selection

Two authors (E.L., D.B.) independently reviewed the titles and abstracts of all papers retrieved by the search strategy. Relevant full-text articles were evaluated for fulfilling the inclusion and exclusion criteria.

### 2.4. Quality Assessment

The evaluation of case-control studies was carried out according to the Newcastle–Ottawa Scale (NOS), which is an effective tool for quality assessment. The estimation of the three categories was performed as follows: selection (adequate case definition, representativeness of the cases, selection of controls, definition of controls); comparability (factors that the study controlled for by design or analysis to improve the comparability of baseline characteristics of cases and controls); exposure (ascertainment of exposure, same method of ascertainment for cases and controls, nonresponse rate).

### 2.5. Data Extraction

Two authors (E.L., D.B.) extracted the following data from the eligible studies in an independent manner: first author, publication year, country where the study was conducted, participant characteristics (sample size, age, sex, current treatment), sample materials, DNA extraction method, 16S rRNA variable region, sequencing platform, data analysis platform, reference sequences database employed in the studies, microbial diversity, alterations in the skin and gut microbiota in patients with HS.

## 3. Results

### 3.1. Search Results and Study Characteristics

Twenty-six records were included for the final investigation. The details of the article selection process are shown in Figure 1.

All eligible studies were published between 1996 and 2021 [11,12,13,14,15,16,17,18,19,20,21,22,23,24,25,26,27,28,29,30,31,32,33,34,35,36]. The total sample size of included studies was 1628 (1419 patients with HS and 209 healthy controls).

The five tables were constructed including the main findings from each incorporated study (Table 1, Table 2, Table 3, Table 4 and Table 5).

Twenty-five studies examining the cutaneous microbiome and five studies examining the gastrointestinal microbiome in HS were identified. A total of 1274 patients and 123 healthy controls were included across studies examining the cutaneous microbiome. A total of 145 patients and 86 controls were included across gut microbiome studies.

### 3.2. The Cutaneous Microbiome in Hidradenitis Suppurativa

See Table 1, Table 2, Table 3 and Table 4.

### 3.3. The Gastrointestinal Microbiome

See Table 5.

### 3.4. The Oral Microbiome

No studies were identified which examined the oral microbiome in HS.

### 3.5. Quality of the Evidence

All included studies were assessed for quality according to NOS and scored between 2 to 9 points (Table 1, Table 2, Table 3, Table 4 and Table 5).

## 4. Discussion

### 4.1. The Human Microbiota and Skin Microbiome

The human microbiota refers to microorganisms found on humans. It is composed of communities of bacteria, viruses, and fungi and is known to have a greater complexity than the human genome itself. The collective microbiota together with its inhabitation of a certain environment is called microbiome. The composition of the human microbiome suggests that the most of it consists of the commensal microorganism, predominantly bacteria, and the disruption of this changeable homeostasis could lead to dysbiosis. Considering that the microbiome balance is crucial for the immune system development, any unevenness can contribute to disruption in communication between the microbiome and the host [37]. The skin microbiome is constantly influenced by various exogenous and environmental factors, such as skin-care products, clothing, humidity, temperature of the surrounding climate, and ultraviolet light exposure. Moreover, certain studies demonstrate that there is a large variability in the density and types of bacteria forming the microbiome, dependent on characteristics such as age, sexual maturity, anatomical location (i.e., face, trunk, extremities), skin pH, personal hygiene, and topography of the area, including hair, sebaceous glands, and moisture [5,6,38,39].

### 4.2. Gut–Skin Axis

In 1907, Metchnikoff hypothesized that overall health and longevity are intimately linked to the gut microbiota [40]. Considering the huge immunological impact and metabolic capacity of the gut microbiota, it is postulated that the gut microbiota is the essential part of the gut–skin axis (GSA). In this particular connection, the gut can enhance skin health via modulation of the immune environment through the microbiota. Regardless of multiple studies support the existence of GSA, the causative nature of the relationship between the gut microbiome and dermatologic conditions still is not elucidated. For instance, almost one-fourth of patients with conditions primarily affecting the gut also suffers from associated cutaneous findings, mainly psoriasis and cutaneous ulcers [41,42].

Various hypotheses propose a common argument suggesting that inflammation, originating from intestinal dysbiosis in the gut or skin, may trigger negative effects on the skin through the GSA [43]. Additionally, a theory indicates that bacteria migrate directly from the gut to other organs, and circulation occurs due to increased intestinal permeability [44]. Multiple studies have found an elevated presence of gut bacterial DNA in the bloodstream of patients with chronic skin conditions, which contributes to the inflammatory response [45]. The neuroendocrine link between the gut and skin microbiome is facilitated by gut microorganisms producing neurotransmitters such as norepinephrine, serotonin, and acetylcholine. This leads to the stimulation of neural pathways, resulting in the release of hormones from enteroendocrine cells, causing widespread systemic effects and inflammation, ultimately affecting the skin.

### 4.3. Methodology

A remarkable increase in research on the microbiome in recent years can be attributed to advancements in the field’s methodology. Two major sequencing-based techniques used for microbiome research are metataxonomics and metagenomics [46]. The first one involves sequencing specific marker genes, such as 16S rRNA or the internal transcribed spacer 1 (ITS1), that have hypervariable regions for identifying species and conserved regions for targeting with universal primers [47]. This method is quick and cost-effective, but it cannot detect viral sequences and has limited resolution. On the other hand, metagenomics involves sequencing all genetic material in a sample without prior locus selection, allowing for de novo genome assembly and high-resolution analysis of species and strains. However, it requires a high read count and alignment to a reference genome for classification and may also capture reads from the host organism. Multi-omics approaches, including transcriptomic, proteomic, and metabolomic data, can supplement sequencing data for a more comprehensive understanding of microbial communities [48]. As these techniques continue to develop, they hold promise for advancing biomarker discovery and treatment development for various health conditions.

As the integration of multi-dimensional datasets becomes more sophisticated, our understanding of microbiota biology in relation to health and disease is expected to improve, which could lead to advancements in biomarker discovery or the development of new treatments.

### 4.4. Research

The majority of the studies presented in Table 1 found a decrease in the diversity of the microbiome in HS patients compared to healthy controls. For example, Schneider et al. (2017) [22] reported a significant loss of variety in HS patients compared to healthy controls, and Guet-Revillet et al. (2017) [24] found that anaerobes were increased in lesional skin of HS patients.

Other studies found differences in specific bacterial taxa between HS sufferers and the control group. For instance, Lam et al. (2021) [11] found that *Mesorhizobium* was increased in lesional skin of HS patients compared to healthy controls, and Ring et al. (2017) [19] reported that *Porphyromonas* and *Peptoniphilus* were more abundant in HS skin than healthy skin. Naik et al. (2020) [14] found that *Cutibacterium* spp. were decreased in HS patients, while Gram-negative *Porphyromonadeacea*, *Prevotellaceae*, *Fusobacteria*, and positive anaerobes (*Clostridales*) were increased in the HS group.

Some studies also found that the microbiota in different body sites or skin niches of HS patients are less diverse and more similar to each other than in healthy controls. For example, Schneider et al. (2020) [22] reported a loss of heterogeneity between body sites and skin niches in HS patients. McCarthy et al. (2022) [12] found that *Finegoldia magna* was increased in groin and axilla of HS patients but decreased in nasal swabs of HS patients.

The research that used traditional culture methods (Table 2) also found differences in the bacterial composition of HS patients compared to healthy controls. The major findings include the identification of *Staphylococcus aureus*, *Diphtheroid*, *Escherichia coli*, and coagulase-negative staphylococci as the most common bacteria in HS patients, as well as the correlation between higher Hurley stages and more polymicrobial flora in culture-based studies.

Table 3 presents the results of (PNA)-FISH and CLSM studies on skin microbiota in patients with HS and healthy subjects. The studies utilized punch biopsy samples of clinically unaffected HS skin and employed PNA-FISH and CLSM examinations to assess bacterial aggregates and diversity. The major findings include the absence of bacterial aggregates at the stratum corneum and in hair follicles of preclinical HS skin. Only 12% of HS samples were categorized as positive for bacterial aggregates, while a morphologically significant presence of bacterial aggregates was observed in 92% of the healthy controls.

Table 4 presents the results of FISH, histology, and IF studies on skin microbiota in HS patients compared to healthy controls. The studies used tissue samples to examine bacterial colonization and utilized techniques such as histology and immunofluorescence (IF). The major findings include 63% of the HS patients being positive for bacterial colonization based on histological analysis. DAPI-labeled coccoids were observed in 71% of the positive patients, predominantly in the form of biofilms and microcolonies. Specifically, *P. acnes* was found as biofilms in hair follicles of two patients, while *Staphylococcus aureus* and coagulase-negative staphylococci were not detected in any of the samples.

The results presented in Table 5 summarize the findings from five metagenomic sequencing studies investigating the gut microbiota in both of the studied groups. These studies employed different sample types and sequencing regions and measured α- and β-diversity to explore the differences in the composition and structure of the gut microbiota between the two groups.

Collectively, these findings highlight that HS is associated with alterations in the gut microbiota composition, characterized by changes in specific bacterial species and reduced overall microbial diversity. The presence of certain bacterial species, such as *Robinsoniella peoriensis*, *Bilophila*, *Holdemania*, and *Ruminococcus callidus*, suggests potential roles in the pathogenesis of HS.

### 4.5. Limitations and Correlations

The concept of a “normal” cutaneous microbiome is still in the developmental stage and primarily differs based on the anatomical location of the skin rather than ethnicity or the use of typical topical products. The Human Microbiome Project is currently analyzing the “normal” human microbiome in various anatomical settings, including the skin. However, the existence of a true “normal” microbiome is unlikely, and it is expected to vary within and across individuals over time. Hence, comparing the results with “normal healthy” controls could introduce biases in identifying the findings [49,50].

There are several limitations in microbiome research related to HS. Most studies lack patient demographic data, such as smoking status and body mass index (BMI), which have demonstrated an impact on the intestinal microbiome, causing reduced diversity and alterations in commensal bacteria [51]. Additionally, the methodology employed in these studies varied, including the way specimens were obtained and the identification of bacteria, leading to differences in the bacteria identified on normal skin. Therefore, future studies must include adequate metadata related to age, sex, ethnicity, disease severity, anatomical location, concomitant medication, prior or concurrent antibiotic use, and topical product use to avoid confounding factors [52].

The investigation of microbiome alterations in HS among individuals with distinct body composition phenotypes has shed light on intriguing findings. HS has been associated with obesity, prompting researchers to explore the potential influence of body composition on the microbiome in this condition. This proffers valuable insights into the interplay between HS pathogenesis and the composition of the microbial communities inhabiting the human body.

Research indicates that HS patients with obesity exhibit notable differences in their microbiome compared to nonobese individuals with HS. These differences encompass alterations in both the gut and skin microbiota. Obese HS patients tend to exhibit dysbiosis, characterized by imbalances in microbial diversity, composition, and functionality. It is postulated that these dysbiotic changes in the microbiome could contribute to the chronic inflammatory state and the clinical manifestations observed in obese individuals with HS.

Specific bacterial taxa have been implicated in the altered microbiome of HS patients with obesity. The increased abundance of certain taxa, such as *Staphylococcus* species, has been observed in the skin microbiota of obese HS patients. Furthermore, dysregulation of bacterial species, including *Prevotella* and *Bacteroides*, has been noted in the gut microbiota of these individuals.

Conversely, nonobese HS patients display distinct microbiome characteristics compared to their obese counterparts. While there may still be some degree of dysbiosis, the specific taxa involved and the extent of microbial perturbations may differ. The microbial alterations in nonobese HS patients may be more subtle and nuanced, requiring further investigation to fully elucidate their implications for disease pathogenesis.

Understanding the microbiome changes associated with different body composition phenotypes in HS can have significant implications for patient management [29,53].

The identification of specific microbial signatures associated with HS has raised the possibility of utilizing the microbiome as a biomarker for this condition. By analyzing the composition and relative abundance of key bacterial taxa, it may be possible to develop microbiome-based biomarkers that can aid in the diagnosis and stratification of HS patients. Furthermore, studying the dynamics of the microbiome over time may offer insights into disease progression and treatment response. Additional investigation is warranted to better understand the significance of bacterial-based biomarkers in the context of HS, considering the well-established association between HS and bacterial dysbiosis. Several studies have highlighted the presence of inflammatory markers in HS, and it is acknowledged that the expression of these markers may be influenced by the presence or absence of specific microorganisms. Therefore, it is essential to conduct future studies that correlate tissue and serum inflammatory profiles with microbiome analyses to ascertain the clinical relevance and potential coexistence of bacterial dysbiosis with identified immunological alterations in HS. Such research endeavors will provide valuable insights into the intricate interplay between the microbiome and immune dysregulation in HS [54].

Additionally, the methodology employed in these studies varied, including the way specimens were obtained and the identification of bacteria, leading to differences in the bacteria identified on normal skin. Therefore, future studies must include adequate metadata related to age, sex, ethnicity, disease severity, anatomical location, concomitant medication, prior or concurrent antibiotic use, and topical product use to avoid confounding factors [29].

Cultures are crucial in identifying bacteria, particularly in clinical settings, but they have limitations as it is estimated that 99% of bacteria cannot be grown on culture [55]. The use of 16S is beneficial in identifying an array of bacteria compared to the traditional bacterial culture, but it cannot shed light on bacterial activity, host response, or interactions between host and microbe. Therefore, further research using whole-genome sequencing and RNA transcriptomics responses is necessary to better understand bacterial activity and the interplay between the organism and host gene expression [56]. Furthermore, standardization of approaches to collect affected and unaffected tissue within and across participants and associated baseline patient data is crucial to interpretation of data and minimizing confounding factors [57].

Biofilms and their role in HS pathogenesis are outside the scope of this review article but are still significant triggers in the organism-host pathophysiological response and deserve a mention in the discussion. HS is a disease characterized by biofilm formation, which may be significant in triggering the host pathophysiological response [58].

## 5. Conclusions

The human skin is a unique environment that presents challenges for microbial studies due to its low microbial biomass, high risk of contamination, and diverse range of site-specific microorganisms. To avoid bias, it is crucial for researchers and clinicians to acknowledge that there are several factors that may limit the comparability of microbiome research, such as variations in sampling and methodology between studies, as well as laboratory factors during DNA extraction and downstream processing.

Collectively, the findings demonstrate that HS is associated with dysbiosis in the skin microbiome, characterized by reduced microbial diversity, specific bacterial taxa differences, similarity within body sites, and alterations in the gut microbiota. These insights contribute to our understanding of the potential role of microbiota in the pathogenesis of HS and may guide future research and therapeutic approaches targeting the microbiome.

## Figures and Tables

**Figure 1 biomedicines-11-02277-f001:**
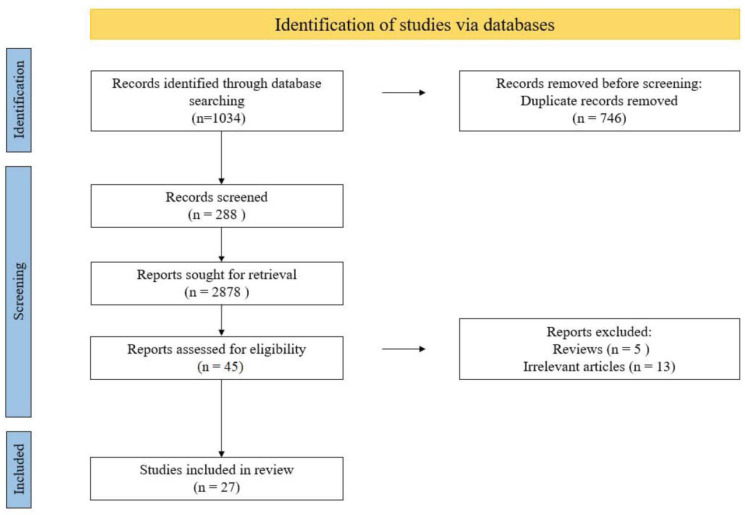
PRISMA flow diagram of study selection for inclusion in the systematic review; PRISMA—Preferred Reporting Items for Systematic Reviews and Meta-Analyses.

**Table 1 biomedicines-11-02277-t001:** Metagenomic sequencing skin studies.

Study ID	# of Patients	# of Healthy Controls	Sample Type	16S Region	α–Diversity	β–Diversity	Major Findings	NOS
Lam 2021 [11]	7	6	swabs	V3–V4	no significant differences	no significant differences	*Mesorhizobium* ↑ in lesional skin	4
Schneider 2020 [22]	11	10	swabs/cyanoacrylate glue	V3–V4	no significant differences	significant loss in HS patients; impact of smoking and alcohol use	loss of heterogeneity between body sites and skin niches in HS patients; *Cutibacterium* ↓; *Peptoniphilus*, *Porphyromonas*, *Agrobacterium*, *Pseudomonas*, and *Arcanobacterium* ↑; functional differences between microbiota of HS and normal skin; metabolic pathways are influenced by different genera in normal and HS skin	8
Guet-Revillet 2017 [24]	65	ND	swabs/biopsy/needle aspiration	V1–V2	ND	ND	anaerobes in lesional skin ↑; *Prevotella* and *Porphyromonas* ↑; *Fusobacterium* and *Parvimonas*—correlation with HS severity	5
McCarthy 2022 [12]	59	20	swabs	ND	reduction in HS cohort (only nasal swabs reached statistically significant decrease)	significant separation in beta-diversity with respect to axilla, groin, and nasal microbiota datasets	*Finegoldia magna* ↑ (groin and axilla) in HS	9
Naik 2020 [14]	12	5	swabs	V1–V3	significant ↑ in the inguinal creases of HS (affected and unaffected skin)	ND	*Cutibacterium* spp. ↓ in HS (affected and unaffected); Gram negative *Porphyromonadeacea*, *Prevotellaceae*, *Fusobacteria phylum* + positive anaerobes (*Clostridales)* ↑ in HS (affected and unaffected skin)	6
Ring 2017 [18]	30	24	biopsy (HS: lesional and non-lesional skin)	V3–V4 + V3–V4 of the 18SrDNA	no significant difference; increased Shannon diversity index was found in non-lesional HS skin compared to lesional and healthy skin	no significant difference	*Corynebacterium* and *Porphyromonas* and *Peptoniphilus*—HS skin; *Acinetobacter + Morxella* ↑ in non-lesional HS; *Porphyromonas* and *Peptoniphilus* spp. at the genus level sig ↑ in lesional skin; *Cutibacterium* spp. ↑ in healthy controls; *P. acnes* and *Corynebacterium striatum* ↑ in the healthy control group	7
Guet-Revillet H 2014 [16]	6	ND	transcutaneous/swabs	ND	ND	ND	*Staphylococcus* ↑ within the Hurley stage 1 abscess; *Prevotellam Porphyromonas*, *Anaerococcus and Mobiluncus* spp. ↑ in chronic suppurating lesions	3
Riverain-Gillet 2020 [17]	60	17	swabs	ND	no significant difference	significantly higher in HS samples	*Prevotella*, *Actinomyces*, *Campylobacter ureolyticus* and *Mobilinucus* ↑ staphylococci ↓ in HS samples; two clusters: Cluster A—enriched in anaerobes (*Prevotella*, *Porphyromonas*, *Dialister and Peptoniphilus asaccharolyticus*), Cluster B—aerobic) associated with *Staphylococcus* and *Micrococcus*	7

ND—not determined.

**Table 2 biomedicines-11-02277-t002:** Culture-based skin studies.

Study ID	# of Patients	# of Healthy Controls	Sample Type	Culture Methods	α–Diversity	β–Diversity	Major Findings	NOS
Jamalpour 2019 [25]	26	ND	swabs	standard culture-based methods	ND	ND	*Staphylococcus aureus*, *Diphtheroid*, and *Escherichia coli*—most common	2
Nikolakis 2017 [26]	50	ND	swabs	5% sheep blood agar—aerobic bacteria/Schaedler agar + 5% sheep blood—anaerobic bacteria	ND	ND	*S. aureus*, *S. epidermidis*, *E. faecalis*, *E. coli*, *P.bivia*, and *P. disiens*—most common; Higher Hurley stages correlated with more polymicrobial flora	3
Thomas 2016 [27]	76	ND	swabs	standard culture-based methods	ND	ND	*Corynebacterium* *Species*, *Staphylococcus epidermidis* and *Staphylococcus aureus*—most common	2
Hessam 2016 [28]	113	ND	deep swabs	standard culture-based methods	ND	ND	coagulase-negative staphylococci and *Staphylococcus aureus*, *Proteus mirabilis* and *Escherichia coli*—most common; low resistance rate for cotrimoxazole	3
Haskin 2016 [29]	189	ND	purulent drainage swab	NC	ND	ND	*Firmicutes* ↑—among obese HS patients vs. nonobese	2
Matusiak 2014 [31]	69	ND	swabs	standard culture-based methods	ND	ND	*S. epidermidis*, *Proteus mirabilis*, *S. aureus*, *Enterococcus faecalis*—most common; carbapanems, penicillins with β-lactamase inhibitors and fluoroquinolones—highest effectiveness	3
Sartorius 2012 [32]	10	ND	swabs (superficial and deep)/skin biopsy	standard culture-based methods	ND	ND	*Staphylococcus aureus* not found in any lesions; coagulase-negative Staphylococci, *Corynebacterium* spp.—most common	2
Lapins 1999 [33]	25	ND	swabs (superficial, middle and deep)/skin biopsy	standard culture-based methods	ND	ND	*Staphylococcus aureus*, coagulase-negative staphylococci, *Peptostreptococcus* spp., *Cutibacterium acnes*, *S. aureus*—most common	3
Brook 1999 [34]	17	ND	swabs	standard culture-based methods	ND	ND	*S. aureus*, *Streptococcus pyogenes*, *Pseudomonas* *Aeruginosa*, *Peptostreptococcus* spp., *Prevotella* spp., micro-aerophilic streptococci, *Fusubacteriurn* spp., *Bacteroides* spp.—most common	2
Jemec 1996 [35]	41	ND	pus aspiration	standard culture-based methods	ND	ND	*S. aureus*, *S. milleri*, *S. epidermidis*, *S. hominis*, *Corynebacterium* spp., *Acinetobacter* spp., *Lactobacillus* spp.—most common	3
Katoulis 2015 [15]	22	ND	direct percutaneous needle aspiration of abscess	sheep blood (5%), chocolate and MacConkey agar plates were incubated under anaerobic conditions.	ND	ND	7 were culture negative and 15 culture positive; 16 isolates obtained (14 aerobic, 2 anaerobic); *P. mirabilis*, *Staphylococcus haemolyticus* and *Staphylococcus lugdunensis*—predominant aerobic species; *Dermacoccus nishinomiyaensis* and *Cutibacterium granulosum*—isolated anaerobic bacteria	3
Guet-Revillet H 2014 [16]	82	ND	transcutaneous; swabs	anaerobic bacteria growth—homogenization of the biopsy samples using a sterile porcelain mortar in 0.5 mL of Schaedler broth, purulent drainage and swab specimens discharged in 0.5 mL of Schaedler broth, Uriselect4 agar plate, a colistin-nalidixic acid (CNA) blood agar plate, and a Columbia blood agar plate.	ND	ND	106 out of 126 lesional samples positive; two microbiological profiles detected: Profile A—*Staphylococcus lugdunensis* as a unique or predominant isolate, Profile B—a mixed anaerobic flora of strict anaerobes and/or anaerobic actinomycetes and/or streptococci of the milleri group	3
Riverain-Gillet 2020 [17]	60	17	ND	NC	ND	ND	*S. epidermidis*, *Staphylococcis hominis*, *Cutibacterium avidum* and *Cutibacterinum acnes* ↑ in the skinfolds of HS subject; mean abundance of anaerobes ↑ in Hs skinfolds	7
Benzecry 2018 [21]	46	ND	swabs	chocolate agar, Columbia agar with 5% sheep blood, Mannitol salt agar, MacConkey agar, Schaedler CNA agar with 5% sheep blood, Schaedler neomycin—vancomycin with 5% sheep blood agar, Columbia CNA agar with 5% sheep blood.	ND	ND	31 cultures (52%) positive; total of 15 bacterial species isolated: nine aerobes and six anaerobes; *Enterobacteriaceae* the most frequent isolates (#11 = 35%), followed by *Streptococcus* spp. (#8 = 26%), *Corynebacterium* spp. (#7 = 23%) and *Staphylococcus* spp. (#6 = 19%).	3

ND—not determined, NC—not clear.

**Table 3 biomedicines-11-02277-t003:** (PNA)-FISH and CLSM skin studies.

Study ID	# of Patients	# of Healthy Controls	Sample Type	Method	α–Diversity	β–Diversity	Major Findings	NOS
Ring 2017 [19]	24	24	punch biopsy (clinically unaffected HS skin)	PNA-FISH and CLSM examinations	ND	ND	absence of bacterial aggregates at the stratum corneum and in hair follicle—preclinical HS skin; 12% of HS samples categorized as positive; morphologically significant presence of bacterial aggregates in 92% of the healthy controls	7

ND—not determined.

**Table 4 biomedicines-11-02277-t004:** FISH, histology and IF (Immunofluorescence of the skin) skin studies.

Study ID	# of Patients	# of Healthy Controls	Sample Type	α–Diversity	β–Diversity	Major Findings	NOS
Jahns AC 2014 [23]	37	ND	tissue samples	ND	ND	histology: 17 patients (63%) positive for bacterial colonization; DAPI labeled coccoids seen in 71% of the positive patients in the form of biofilms and microcolonies; *P. acnes* as biofilms in hair follicles of two patients; *Staphylococcus aureus* and coagulase-negative staphylococci not detected in any sample	2

ND—not determined.

**Table 5 biomedicines-11-02277-t005:** Metagenomic sequencing studies gut.

Study ID	# of Patients	# of Healthy Controls	Sample Type	16S Region	α–Diversity	β–Diversity	Major Findings	NOS
Lam 2021 [11]	17	20	stool	V3–V4	no significant differences	no significant differences	*Robinsoniella peoriensis* in HS patients; *Sellimonas* ↑; *Christensenellaceae* ↓	4
S. Kam 2020 [36]	3	3	fecal samples from the central portion of the specimen	ND	no significant differences (measured with operational taxonomic unit (OUT)); significant difference: greater in the control group (measured with Shannon index)	no significant differences (measured with weighted UniFrac distance matrices)	HS patients: *Firmicutes* ↓ *Bilophila* + *Holdemania* ↑ *Lachnobacterium* + *Veillonella* ↓	8
Eppinga H 2016 [13]	HS only = 17 concomitant HS and IBD = 17	33	20 mg of feces	ND	no significant difference	NC	*F. prausnitzii* ↓ patients with IBD + HS; no significant difference between the abundance of *E. coli* and HS cohorts and healthy controls	8
McCarthy 2022 [12]	59	30	fecal	ND	significantly ↓ in HS	less clustering within the HS samples	*Ruminococcus callidus + Eubacterium rectale* ↑ in HS; the greatest amplicon sequence variants (ASVs) assigned to the taxa *Streptococcus* spp. and *Ruminococcus gnavus*	9
Ring 2019 [20]	32	ND	skin covering tunnels	V3–V4 + V3–V4 18S rDNA	ND	ND	*Porphyromonas* spp. and *Prevotella* spp. ↑ gelatinous material in the HS tracts; *Corynebacterium*, *Staphylococcus*, and *Peptoniphilus* ↑	3

ND—not determined, NC—not clear.

## Data Availability

Publicly available datasets were analyzed in this study. This data can be found here: https://www.ncbi.nlm.nih.gov/pmc/.

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
