# Peer review of "Skin and Gut Microbiome in Hidradenitis Suppurativa: A Systematic Review"

_biomedicines, 2023, doi:10.3390/biomedicines11082277_

Round 1

Reviewer 1 Report

The authors present a systematic review about the current knowledge in bacteriology of HS. The paper is well written, methods are well described. I think a paper reporting a microbiological study of HS samples, lymph notes and fistulas should be added in the table with cultural tests (Vaienti S, et al. J clin med).

Unfortunately this study does not add anything new in the literature about HS until the results of microbiome project will be available. Nevertheless I appreciate the work of the authors.

Author Response

Dear Reviewer,

Thank you for the revision. The appropriate changes were made - the mentioned article was added to the Table 1 and the additional changes were made in the manuscript.

Reviewer 2 Report

The authors of this manuscript carry out a systematic review of the gut and skin microbiota in HS sufferers.

The topic is a "hot" HS subject and has been reviewed ; PMID: 36534543 PMID: 36232581 PMID: 36116091 PMID: 34696185

To distinguish themselves from the other reviews, it would make reasonable sense for the authors to discuss possible;

> changes in microbiome in different HS with different body composition phenotypes ex;

Haskin A, Fischer AH, Okoye GA. Prevalence of Firmicutes in Lesions of Hidradenitis Suppurativa in Obese Patients. JAMA Dermatol. 2016;152(11):1276-1278. doi:10.1001/jamadermatol.2016.2337

Bervoets L, Van Hoorenbeeck K, Kortleven I, et al. Differences in gut microbiota composition between obese and lean children: a cross-sectional study. Gut Pathog. 2013;5:10. doi:10.1186/1757-4749-5-10

> whether the a distinct microbiome is present in patients with a different HS phenotype

> whether the microbiome can act as a biomarker (already desginate GRADE moderate/severe diagnostic biomarker PMID: 35044423, can it behave as a prognostic or therapeutic marker)

The grammar and synthax need revision.

Author Response

Dear Reviewer,

thank you for your comments and suggestions. The recommended changes were introduced into the manuscript.

The changed lines 237-269 are highlighted in the text.

Reviewer 3 Report

The paper discusses the microbiome, mostly gut, in patients with HS, a disease that is known to negatively impact mental health and life quality when at the same time, no good treatments exist. The authors have reviewed a large amount of literature.

Investing the role of gut and skin microbiome on the pathogenesis and aetiology of HS is progressing quickly, and the authors present relevant studies on this issue in an attempt to clarify this role.

The methodology is sound and the results are clearly presented. The discussion and conclusions are well written, although, at the present point, much still needs to be explored. The authors prompt to more research in this area.

My only comment is on some words being written in red. What is the meaning of those?

Some typos/grammar mistakes need to be checked under proofreading. E.g. P. acnes is mentioned. It is now C. acnes and is given with both versions. Perhaps this is because the study was older, but it might be important to explain this since younger readers may not be aware that those are the same bacterium.

Author Response

Dear Reviewer,

Thank you for the revision. The fields in red were highlighted due to the corrections because of the repetitions. Thank you for noticing the bacteria taxa name differences. The changes were introduced into the manuscript.